# Assessment of Waste Glass Incorporation in Asphalt Concrete for Surface Layer Construction

**DOI:** 10.3390/ma16144938

**Published:** 2023-07-11

**Authors:** Stavros Kalampokis, Danai Kalama, Fotini Kesikidou, Maria Stefanidou, Evangelos Manthos

**Affiliations:** 1Highway Engineering Laboratory, School of Civil Engineering, Aristotle University of Thessaloniki, 54124 Thessaloniki, Greece; skalampokis@civil.auth.gr (S.K.); danaimark@civil.auth.gr (D.K.); 2Laboratory of Building Materials, School of Civil Engineering, Aristotle University of Thessaloniki, 54124 Thessaloniki, Greece; kesikidou@civil.auth.gr (F.K.); stefan@civil.auth.gr (M.S.)

**Keywords:** Glassphalt, waste glass, stiffness, cyclic compression test, ITSR, solar reflectance index

## Abstract

The growing need to preserve natural resources and minimize landfill waste has led to an increased consideration of incorporating waste materials in road construction and maintenance. This study focuses specifically on utilizing waste glass as part of the aggregates in hot asphalt, particularly in Asphalt Concrete (AC) for surface layers, known as “Glassphalt”. Glass, due to its poor adhesion to bitumen, presents challenges when used in asphalt mixtures. Two types of waste glass, monolithic and tempered, were incorporated at two distinct contents, 10% and 15%, into the AC. Several properties such as stiffness, resistance to permanent deformation (evaluated through cyclic compression tests), indirect tensile strength, and the indirect tensile strength ratio (ITSR) were assessed for all Glassphalt mixtures, as well as the conventional mixture. Additionally, the Solar Reflectance Index (SRI) was measured to evaluate the reflectivity of the resulting Glassphalts. The findings indicate that the incorporation of both types of waste glass resulted in reduced stiffness and resistance to permanent deformation. Regarding water sensitivity (ITSR), the Glassphalts containing 15% waste glass, regardless of the glass type, exhibited ITSR values below the accepted threshold of 80%. The addition of waste glass did not yield significant changes in SRI measurements.

## 1. Introduction

Waste glass is a major environmental issue due to its slow decomposition rate and significant volume generated annually. In recent years, the reuse of waste glass in construction materials has gained significant attention as a sustainable solution. One such application is the use of waste glass in asphalt mixtures for pavement construction. The integration of waste glass in asphalt mixtures is an effective approach to reducing the waste stream and promoting sustainable development.

The use of waste glass in asphalt mixtures is dependent on various factors, such as the type of waste glass, particle size, and glass content. The engineering properties of asphalt mixtures containing waste glass are influenced by these factors, and their optimization is crucial for achieving the desired performance.

In summary, the use of waste glass in asphalt mixtures is a promising approach to sustainable construction, reducing the waste stream, while promoting the use of recycled materials in pavement engineering. This paper aims to investigate the use of two types of waste glass in two distinct contents in asphalt concrete for surface layers and evaluate the performance of the latter.

## 2. Background

To optimize the performance of building materials, engineers must take into account the principles of sustainability and cyclic economy. This involves considering the utilization of various waste materials generated from manufacturing operations, service industries, sewage treatment plants, households, and mining. Glass, an inorganic and non-metallic material, is produced through the sintering of carefully selected raw materials. Due to its properties, glass cannot be effectively incinerated or decomposed. However, recycling glass offers a solution that conserves energy and reduces environmental waste. The focus on glass recycling technology is crucial for expanding the utilization of waste glass and promoting the further development of glass recycling techniques. In line with this objective, the European Union has set a target for its Member States to achieve a minimum of 65% recycling or preparation for re-use of glass packaging waste by 2035, as outlined in Directive (EU) 2018/851 [1].

One potential application of waste glass is its utilization in the production of construction materials, specifically incorporating it into asphalt mixtures, which results in a composite material known as Glassphalt [2,3,4]. This incorporation involves replacing a portion of the aggregates in the mixture with waste glass. Numerous studies have been conducted to investigate the performance characteristics of Glassphalt. Two key parameters affecting Glassphalt performance are the glass content and the maximum size of the glass particles used in the mixture. A study conducted by Airey et al. [5] demonstrated that incorporating 10–15% waste glass, along with 2% lime as an anti-stripping agent, and utilizing a maximum glass particle size of 4.75 mm, yielded satisfactory results with reduced risks of tire puncture and skin cutting. Another study by Androjić and Dimter [6] found that the quantity of glass cullet used as aggregates and filler had a significant impact on various characteristics of the mixture, such as Marshal stability, air voids, and density. However, they observed that the performance diminished with a higher glass content.

Regarding the surface course mixtures, Huang et al. [7] reported that asphalt pavements containing 10–15% crushed glass exhibited satisfactory performance, with the maximum permissible size of crushed glass being 4.75 mm due to safety considerations. Higher amounts of larger glass particles in the asphalt mixture may result in inadequate friction and bonding strength, making it more suitable for use in lower courses. To enhance the resistance against stripping, the addition of an anti-strip agent, such as a 2% hydrated lime, is recommended. Importantly, the same manufacturing equipment and paving methods used for conventional asphalt can be employed for producing asphalt mixtures incorporating recycled glass [7].

The mechanical performance of Glassphalts, which directly impacts pavement design, holds significant importance. Arabani et al. conducted a study examining the influence of glass cullet on the stiffness and internal friction of asphalt mixtures [8,9]. The findings revealed that the crushed structure and high angularity of glass particles contributed to an increase in both parameters. Furthermore, Arabani et al. investigated the linear viscoelastic performance of asphalt mixtures incorporating glass particles under varying stress levels and temperatures [10]. The results indicated lower permanent strain and higher thermal sensitivity compared to conventional mixtures. The inclusion of glass particles in asphalt mixtures has also the potential to enhance fatigue performance due to the strength characteristics of the additive materials [11].

A comprehensive investigation was conducted to examine the dynamic characteristics of asphalt mixtures incorporating waste glass in comparison to conventional asphalt concrete mixtures [2]. The study revealed a notable increase in the stiffness modulus of Glassphalts compared to conventional asphalt mixtures. To mitigate stripping-related issues, a 4% hydrated lime additive was employed as an anti-stripping agent, and its effectiveness was compared to a control specimen without any anti-stripping treatment. The glassphalt mixtures with hydrated lime exhibited a substantial improvement in stiffness modulus compared to the other specimens [2]. The recycling of glass materials presents a valuable opportunity to utilize existing resources while reducing environmental impact.

In engineering practice, the behavior of materials used in structures holds significant importance. Consequently, it is essential to investigate the efficiency and behavior of waste materials to determine whether they are superior to or at least equivalent to the currently employed materials [9]. In this regard, a study demonstrated that the increased internal friction resulting from the angularity of glass particles plays a vital role in enhancing the stiffness modulus of specimens with glass cullet content. However, the high smoothness of crushed glass particles hinders adequate bitumen absorption, resulting in a decreased stiffness modulus beyond a certain limit of glass content [12].

Issues related to stripping and moisture damage have also been reported in field projects conducted in New York, Baltimore, and other studies [13]. The smooth surface of glass particles poses a significant challenge as it can disrupt the cohesion between bitumen and stone aggregates, resulting in the stripping of asphalt pavements. To mitigate this problem, additives such as hydrated lime are frequently employed to minimize the detrimental effects of glass–asphalt mixtures while preserving their beneficial properties [14]. Consequently, the incorporation of any material that could potentially exacerbate moisture damage may be deemed undesirable.

Kiletico et al. [15] conducted a study demonstrating that the incorporation of glass powder in asphalt roof shingles increased the solar reflectance index. Additionally, Du et al. [16] suggested that the utilization of glass microspheres as an alternative to limestone filler in asphalt mixtures could lead to a significant reduction in thermal conductivity, up to 24%, thereby enhancing pavement cooling. At the bituminous mastic level, the impact of glass powder was even more pronounced, with a 40% reduction in thermal conductivity and a 60% increase in infrared reflectance [17].

Glass powder has been identified as a potential enhancer for resistance against permanent deformation in paving materials. In a study investigating anti-rutting performance, Du et al. [18] observed an 8.5% reduction in the average rutting depth when glass microspheres were incorporated, compared to an equivalent mixture with limestone mineral filler. Al-Khateeb et al. [19] evaluated the shear properties of asphalt mastics containing waste glass across a wide range of temperatures and loading frequencies. Their findings indicated that adding glass into the asphalt mastic improved stiffness, fatigue resistance, and resistance to permanent deformation, owing to notable enhancements in viscoelastic properties.

Regarding the structural characteristics of fillers, the roundness index, Rigden void, and specific surface area of filler particles were identified as predominant factors influencing the degree of stiffening in mastic [20]. On the other hand, angularity, porosity, aspect ratio, average diameter, and fractal dimension significantly influenced specific rheological properties, including fatigue failure and per cent recovery [21].

The characteristics of filler particle surface and specific surface area played a crucial role in determining bitumen adsorption, which subsequently influenced the rheological and chemical behavior of mastics [22]. Hesami et al. [23] conducted a study revealing that angular geometry had a greater influence on restricting the free movement and rotation of filler particles within the asphalt mastic compared to spherical geometry, resulting in a higher viscosity grade.

The utilization of waste glass in bituminous materials has been recognized for its potential cost-saving benefits and extensive engineering properties. Min et al. [24] proposed that incorporating glass waste into epoxy-asphalt concrete could enhance motorist safety by improving antifriction and reflective properties. In a performance-related study, Simone et al. [25] demonstrated a significant enhancement in the bearing capacity and rutting resistance of flexible pavements by introducing glass powder as a filler. Previous studies have reported notable improvements in performance properties such as Marshall stability, flow, compressive strength, indirect tensile strength, and indirect tensile stiffness modulus [6,26]. However, it is important to note that a high content of recycled glass in asphalt concrete may lead to a decrease in mixture strength, density, voids filled with bitumen (VFA), and void content [27].

Furthermore, investigations have assessed the feasibility of utilizing waste glass in pavement base and subbase courses. It has been found that incorporating glass cullet into recycled concrete aggregates in the subbase layer provides satisfactory shear strength, bearing capacity, and crushing resistance [28,29]. The combination of crushed glass with construction and demolition waste and crumb rubber has also shown a synergistic effect on the unconfined compression strength, California bearing ratio, and resilient modulus of base and subbase layers [30].

In the current study, two types of waste glass are incorporated at two distinct contents into an Asphalt Concrete for surface layers. Properties such as stiffness, permanent deformation (cyclic compression test), indirect tensile strength and indirect tensile strength ratio (ITSR) were determined for all the resulting Glassphalts and the conventional mixture. Solar Reflectance Index was also measured for all asphalt mixtures produced. Based on the results obtained, it was possible to depict the particularities of the Glassphalt concerning the type of waste glass used, as well as the effects induced by the different glass contents.

## 3. Materials and Methods

In the current study, and for the production of the conventional/reference mixture and the Glassphalt with different types and waste glass content, the materials used were a conventional 50/70 bitumen, two types of waste glass (monolithic and tempered) at 10% and 15% contents, and diabase coarse aggregates as long as limestone sand and filler.

### 3.1. 50/70 Bitumen

The characteristic properties of the bitumen used are shown in Table 1.

### 3.2. Waste Glass

The two types of waste glass were supplied in the form shown in Figure 1. In order for these two types of waste glass to be used in asphalt mixture production, a crushing of the materials was conducted. The optical result of the latter procedure as long as the crusher is used is shown in Figure 2.

Figure 3 shows two glass particles (one from each type) under an optical microscope (dino-lite digital microscope, magnification 20×). As shown in Figure 3, the tempered glass particles show a more “cubical” form and are more angular and less elongated than those of the monolithic waste glass.

#### Glass Particle Size Distribution and Glass Performance under Freeze–Thaw Cycles

After crushing, and in order for the two waste glass types to be incorporated into the Glassphalt mixtures, their particle size distribution was determined. The glass portion needed for determining the glass gradations was obtained following EN-933-2-Methods for reducing laboratory samples [36]. The result is shown in Figure 4.

As can be seen from Figure 4, the maximum nominal size (90–10% passing) of both glass types was 10 mm.

Glass properties related to the asphalt mixture design were also determined and the results are shown in Table 2.

As seen from Table 2, monolithic glass particles are more flaky and have fewer cubical particles (considering that the shape index takes into account the non-cubical particles), which confirms the optical result of Figure 3. Freeze–thaw performance was also evaluated for the two types of crushed waste glass at the fraction 4.0–8.0 mm as per procedure defined by EN 1367-01 [39] and EN 1367-06 [40] (presence of NaCl). The mass changes of both waste glass types are ≤0.5%. The Micro–Deval test showed low resistance to wear for both glass types.

### 3.3. Aggregates

As mentioned above, the virgin aggregates used in this study were coarse diabase aggregates and limestone sand and filler. All materials came from quarries near the city of Thessaloniki, Greece.

#### 3.3.1. Aggregate Properties

The properties of coarse diabase aggregates and limestone sand are shown in Table 3.

#### 3.3.2. Coarse Aggregates and Sand Particle Size Distribution

Figure 5 shows the particle size distribution of the coarse aggregates and limestone sand.

### 3.4. Reference Asphalt Mix-Glassphalts

The type of asphalt mix used in this study was a semi-open Asphalt Concrete for surface layers with a nominal aggregate size of 12.5 mm (AC-Type 2) which is the most common asphalt mix type currently used in Greece, for surface layers. The gradation limits for the specific asphalt mix type are shown in Table 4. Five mixes have been produced for the current study: A reference AC-Type 2 (AC Type2-Ref.) with diabase coarse aggregates and limestone sand and filler, two Glassphalt AC-Type 2 with the conventional materials of the reference mix with the addition of 10% and 15% of monolithic glass (AC Type2–10%-Mon. and AC Type2–15%-Mon.), and two Glassphalt AC-Type 2 with the conventional materials of the reference mix with the addition of 10% and 15% of tempered glass (AC Type2–10%-Tem. and AC Type2–15%-Tem.).

### 3.5. Asphalt Mix Design

The asphalt mix design was carried out and the optimum gradation for each case was determined. The optimum bitumen content was determined and was found to be 4.76% per weight of asphalt mix or 5.0% per weight of aggregates for the conventional mix. This result is consistent with the common practice for these mixtures in Greece. In all the Glassphalt mixtures produced, the same bitumen content was used without any adjustment due to the presence of glass. Table 5 shows the respective contents of the aggregate fractions and those of glass for all gradations produced, while Figure 6 shows the optimum aggregate gradations for all cases.

As Figure 6 shows, the optimum gradations, in all cases, are within the specified limits having minimum differences. Hence, an almost uniform gradation has been achieved for all mixtures to be produced.

## 4. Results

In order to investigate the effect of monolithic and tempered glass addition, the following properties have been determined for all mixtures produced: Compactibility, Stiffness, Indirect Tensile Strength (ITS), Water sensitivity (ITSR) and Reflectance. The first four properties cover major mechanical properties and properties related to defects appearing in the presence of water. The reflectance of the resulting Glassphalts is related to the albedo effect and their potential use in urban areas for reducing the heat island effect.

### 4.1. Production of Test Specimens and Slabs

The production of the appropriate number of specimens and slabs for all tests mentioned above, has been carried out using the equipment of the Highway Engineering Laboratory of AUTh. Fifty cylindrical specimens of a 100 mm diameter and ten slabs of 30 × 30 × 4 cm were produced. Table 6 shows the number of specimens and slabs used in each test. The cylindrical specimens were produced by a gyratory compactor and the slabs by a roller compactor. The cylindrical specimens for the compactibility testing were compacted to 200 gyrations, with respect to the relevant EN specification. All other cylindrical specimens were compacted to the same appropriate number of gyrations to achieve the target mix design voids (8–12%). Specimens for stiffness testing, which is a non-destructive test method were also utilized for the ITS and ITSR testing.

### 4.2. Void Content of Specimens Produced

Table 7 shows the void content of the cylindrical specimens produced. Table 7 does not include the specimens used for the determination of the compactibility of the asphalt mixtures, since they were compacted at different numbers of gyrations than the rest of the specimens. As can be seen from Table 7, the air voids of all specimens ranged on average between 10.7% and 11.6%. This range is acceptable by the Greek specifications for AC-Type 2 giving design air void values between 8–12%. The addition of glass has reduced the average voids of the Glassphalt mixtures with respect to the average voids of the AC-T2-Ref. The standard deviation of voids has also been increased, with the addition of glass.

### 4.3. Compactibility

To investigate the compactibility and consequently the effect of glass addition on the workability of the asphalt mixtures produced, the procedure described in the EN 12697-10 [47] standard was followed. According to this standard, asphalt mixture samples are compacted in at least 200 gyrations. The correlation between air voids and gyrations is then graphically illustrated. Gyrations are plotted on the graph in logarithm scale. Finally, an equation of the following form is obtained:u (ng) = u (1) − (K × ln ng)(1)
where: u (ng): void content for a number of gyration ng, (%)

u (1): calculated void content for one gyration, (%)

K: the compactibility

ng: the number of gyrations

Figure 7 shows the compactibility curves for each type of asphalt mixture. The compactibility values were obtained from the average of the corresponding values of two tests for each type of asphalt mixture. Table 8 gives the corresponding equations that describe the relation between air voids/gyrations as long as the parameter K.

Parameter K can be used for the evaluation-ranking of asphalt mixtures based on their compactibility—workability when the initial air voids are the same for all compared asphalt mixtures [52]. In the current study, the initial air voids of the asphalt mixtures are different, and the evaluation will be based on the curves of Figure 7. From Figure 7, it can be seen that for a fixed number of gyrations, asphalt mixtures with glass show a larger number of air voids and are therefore more difficult to compact. Hence, these asphalt mixtures are less workable than the conventional asphalt mixture (for the same compaction temperature and for the specific gradations).

### 4.4. Stiffness

In the current study, the stiffness of asphalt mixtures was determined by the indirect tensile test according to the European standard EN 12697-26 [48]. The test conditions for 100 mm diameter cylindrical specimens were loading time 124 ± 4 ms, target deformation 5 ± 2 µm, and test temperature 20 ± 0.5 °C. To carry out the test, the Nottingham Asphalt Tester (NAT) device was utilized.

Since the test is non-destructive, six specimens were used per type of asphalt mixture, which, at a later stage, were used to determine the indirect tensile strength and water sensitivity of the asphalt mixtures.

Table 9 shows the results of the six specimens per type of asphalt mixture as well as the average and standard deviation of the results.

Figure 8 shows graphically the effect of the glass addition to the AC-T2 mixture and gives the relative reduction in stiffness having as reference the stiffness value of the AC-T2-ref.

As shown in Figure 8, the addition of the two specific glass types reduced the stiffness modulus of the asphalt mix. The greatest reduction was observed with the addition of 15% monolithic glass. The decrease in the stiffness modulus indicates a reduction in the strength of the asphalt mix, and thus, these Glassphalts mixtures should be considered for lower traffic volume roads compared to the traffic volume beard by the conventional mixture.

### 4.5. Indirect Tensile Strength and Water Sensitivity

For asphalt mixtures intended for use in surface layers, the determination of water sensitivity is required to avoid the phenomenon of ravelling. The water sensitivity test should be carried out in accordance with European standards EN 12697-12 [49], procedure A and EN 12697-23 [50]. The specimens to be tested are divided into two groups (usually three specimens per group), based on their average height and average bulk density. In the first group of specimens, the indirect tensile strength at 25 °C shall be determined. The indirect tensile strength shall also be determined for the second group of specimens after they have been placed in a water bath at 40 ± 1 °C for 72 h. At the end of the 72 h, the specimens shall be removed from the 40 °C water bath and placed in a water bath at 25 °C for at least 2 h, completely covered with water. On completion of the above, the indirect tensile strength of the second set of specimens shall be determined. Finally, the indirect tensile strength ratio (ITSR) shall be calculated from:ITSR = 100 × (ITSw/ITSd)(2)
where:

ITSR = Indirect tensile strength ratio, (%)

ITSw = Indirect tensile strength average of a group of specimens, kPa

ITSd = Average indirect tensile strength of the dry test specimen group, kPa

Table 10 gives the results of the tensile strengths of the dry and wet specimens and the respective ITSR per asphalt mixture.

The results of Table 10 show that the addition of both types of glass (monolithic and tempered) reduced the indirect tensile strength of the asphalt mixtures with and without saturation. The addition of 15% monolithic or tempered glass reduced the tensile strength ratio below 80%, making these asphalt mixtures unsuitable for use in surface layers.

### 4.6. Cyclic Compression

The cyclic compression test (CCT) was carried out according to EN 12697-25 method B [51]. The testing parameters are specified by EN 13108-20 [53] for surface courses and are confining stress 150 KPa, amplitude of the axial load 300 kPa, number of load cycles 10,000, and testing temperature 50 °C. The results of cumulative axial strain are shown graphically in Figure 9 and the terms of cumulative axial strain and creep rate are presented in Table 11.

As can be seen from the results in Table 11 and Figure 9, the addition of both glass types increased the cumulative axial strain of the asphalt mix. The Glassphalt with 15% monolithic glass had the highest value of cumulative axial strain, while the conventional bituminous mix had the lowest.

Regarding the creep rate, the addition of glass increased its value in all cases. The Glassphalt with 15% monolithic glass showed the highest rate, while the conventional bituminous mix showed the lowest. Between the two types of Glassphalts, the values of cumulative axial strain and creep rate for the asphalt mixtures with monolithic glass were higher than those of the asphalt mixtures with tempered glass.

### 4.7. Solar Reflectance Index

In the current study, the Solar Reflectance Index (SRI) of the Glassphalt mixtures produced was also investigated in order to detect if the presence of glass particles will affect the solar radiation absorbed or reflected. The latter, if beneficial, could be an indication for the use of Glassphalts as a measure against the heat island effect in urban areas. This investigation is a complex problem related to both the reflectance of the asphalt mixtures to sunlight and artificial street lighting. The individual SRIs of bitumen aggregates and glass are involved in the outcome of the total SRI of the asphalt mixture [54,55,56].

The Solar Reflectance Index (SRI) is a measure of the solar reflectance and emissivity of materials that can be used to indicate how hot they are likely to become when solar radiation is incident on their surface and thus is related to the Albedo effect. The lower the SRI, the hotter a material will likely become in the sunshine. SRI has a scale from 0 to 100 on which materials that absorb and retain solar radiation (and consequently become hotter in the sunshine) have a lower number, whilst highly reflective materials (which consequently remain cooler in the sunshine) have a higher number. In particular, the SRI represents the surface temperature relative to those of the standard white (SRI = 100) and black (SRI = 0) surfaces, as shown in Equation (3) [32].
SRI = 100 × (T_b_ − T_s_)/(T_b_ − T_w_)(3)
where:

SRI = Solar Reflectance Index, per cent

T_s_ = steady-state surface temperature

T_b_ = steady-state temperature of a black surface

T_w_ = steady-state temperature of a white surface

For the present investigation, slabs were created for all asphalt mixtures tested with a size of 30 × 30 × 4 cm. In order to achieve the exposure of the aggregates and glass, the slabs were subjected to a light abrasion to remove the asphalt surface film, which corresponds to the equivalent effect in the field due to the friction of the vehicle tires with the pavement surface. All measurements were made by a specified laboratory and the procedure followed was the one described at ASTM E1980-11 [57]. Figure 10 shows the produced slabs and the light abrasion made, while Figure 11 and Table 12 show the reflection coefficient and the SRI results, respectively.

Table 12 presents the SR (Solar Reflectance) Index values, which are notably small for all asphalt materials, as expected [54,55,56]. The presence of glass particles did not exert a significant influence on the SRI values, yielding results both higher and lower than those obtained with a conventional AC-T2 mixture. This finding leads to the compelling conclusion that the addition of these waste glass types has minimal impact on the SRI value of the asphalt mixture.

Several potential explanations can account for these results. Firstly, it is possible that the specific glass types used in this study are refractive rather than reflective, thus lacking any substantial effect on the overall SRI value of the asphalt mixture. Secondly, the scattering of glass particles within the volume of the slabs may have resulted in their low concentration at the surface, causing the SR values to be primarily influenced by the bitumen, greenish diabase aggregates, and white limestone sand. Lastly, it is plausible that the light surface abrasion conducted in the laboratory was insufficient to fully expose all the glass particles present.

## 5. Discussion for All Test Results

Upon analyzing the results obtained from tests conducted to determine the stiffness modulus, water sensitivity, resistance to permanent deformation, and solar reflectance, the following can be additionally noted.

Glass particles incorporated within the asphalt mix contribute to its resistance under traffic load conditions. However, due to their almost non-absorbent nature and limited affinity with bitumen, these particles tend to shift or experience minimal friction with the aggregate particles when subjected to load. Consequently, this displacement leads to a decrease in stiffness modulus and indirect tensile strength values. In cases of prolonged loading, such as triaxial cyclic compression, this effect becomes more pronounced, resulting in an increase in cumulative axial strain and creep rate. The affinity of glass particles with bitumen affects the water sensitivity of the mixtures, and in some cases, such as the monolithic waste glass, reduces the ITSR < 80%, which is a worldwide common threshold for surface layer asphalts. With respect to the glass type used, it appears that the less “cubical” and angular form of the monolithic glass particles has an impact on stiffness, permanent deformation, and ITSR when incorporating 15% of glass.

SRI measurements did not show any significant result with the use of waste glass and consequently, the use of Glassphalts in urban areas with the specific glass types and in the contents used in this study, will not be suitable for increasing the albedo effect and thus reducing the heat island effect.

## 6. Conclusions

The addition of waste glass reduced the stiffness modulus of the asphalt mix. A greater reduction was observed with the incorporation of 15% monolithic waste glass.The addition of waste glass reduced the resistance of the asphalt mix to permanent deformation. A greater reduction was observed with the incorporation of 15% monolithic glass.The ITSR of asphalt mixtures with glass was reduced compared to a conventional asphalt mixture.Glassphalt mixtures with 15% monolithic or tempered waste glass are not suitable for use in pavement surface layers with respect to an 80% threshold value for surface asphalt layers.The SRI values of all mixtures investigated herein were very low and thus it is not recommended to use the specific types of waste glass in Glassphalts in urban areas.

In summary, the utilization of these specific types of waste glass in surface asphalt mixtures generally led to a reduction in their overall properties. Nevertheless, it is feasible to incorporate the Glassphalts containing 10% waste glass into the surface layers of pavements intended for lower traffic volumes compared to those designed with conventional mixtures for higher traffic. However, when considering the utilization of the Glassphalts with a 15% waste glass content for surface layers in low-volume roads, it is imperative to employ an appropriate antistripping agent.

## Figures and Tables

**Figure 1 materials-16-04938-f001:**
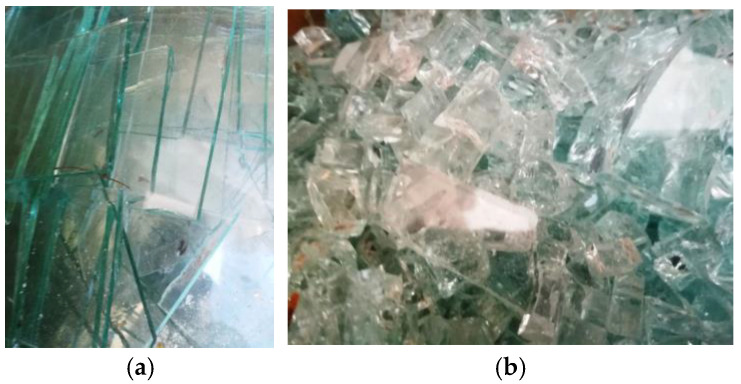
(**a**) Monolithic waste glass before crushing. (**b**) Tempered waste glass before crushing.

**Figure 2 materials-16-04938-f002:**
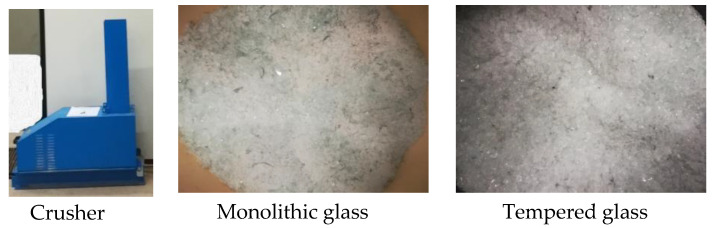
Crusher used and monolithic and tempered waste glass after crushing.

**Figure 3 materials-16-04938-f003:**
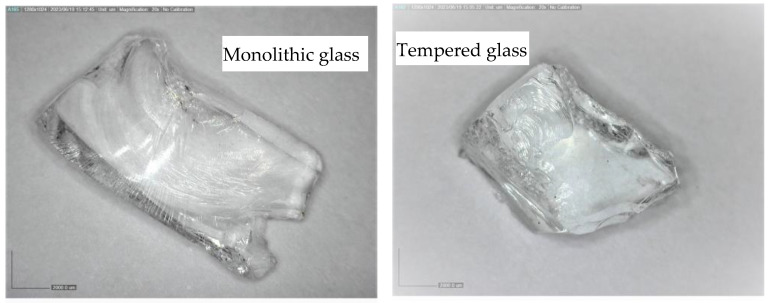
Monolithic and tempered waste glass particles after crushing under an optical microscope.

**Figure 4 materials-16-04938-f004:**
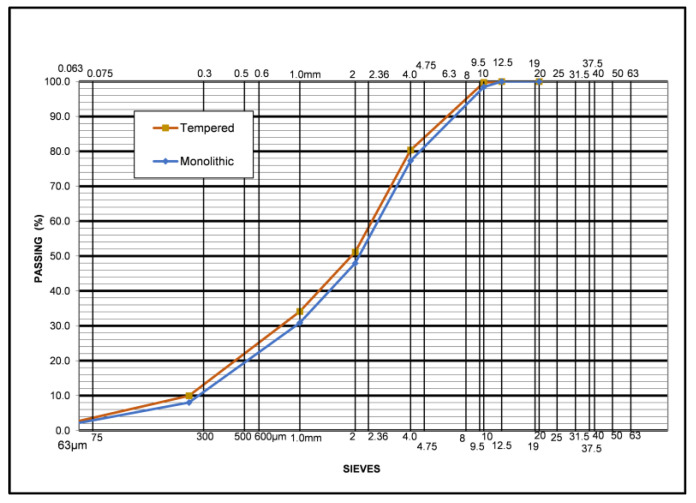
Monolithic and tempered waste glass particle size distribution after crushing.

**Figure 5 materials-16-04938-f005:**
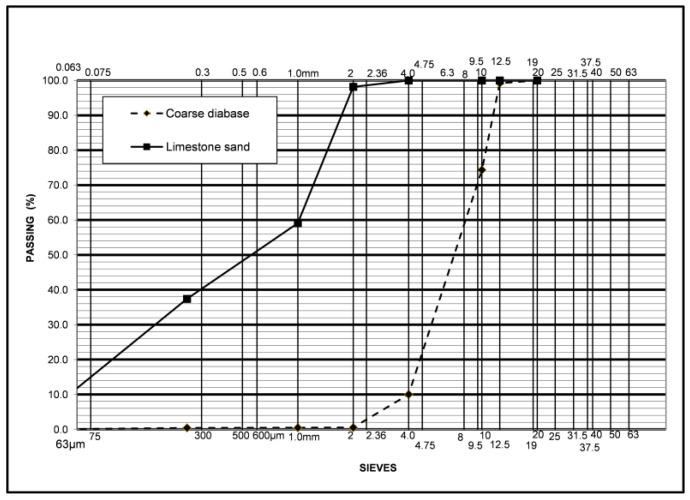
Coarse aggregates and sand particle size distribution.

**Figure 6 materials-16-04938-f006:**
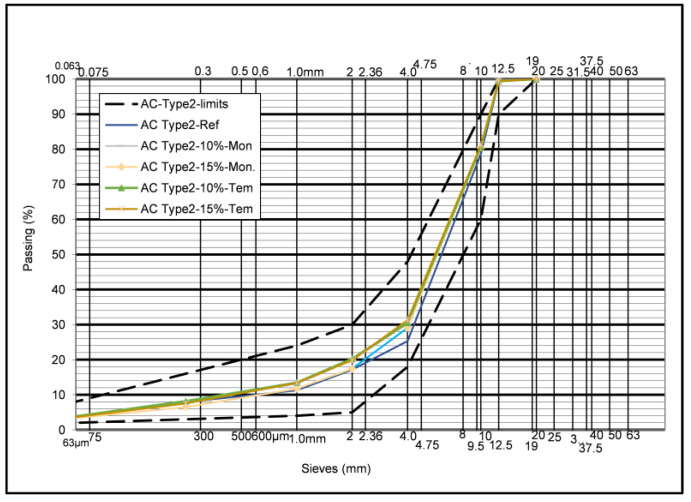
AC-Type 2 particle size distributions.

**Figure 7 materials-16-04938-f007:**
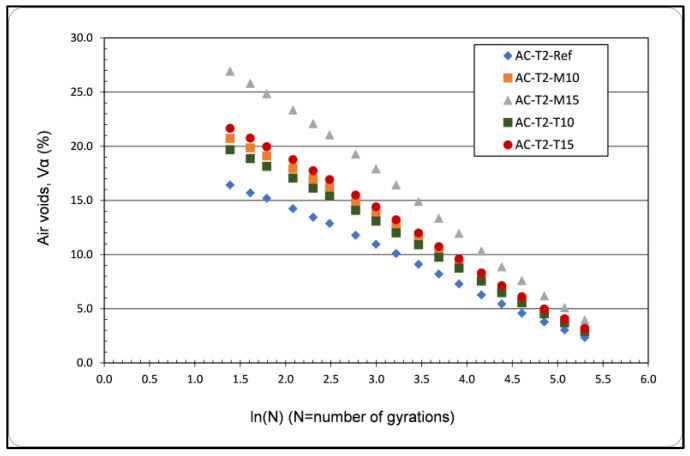
Compactibility curves.

**Figure 8 materials-16-04938-f008:**
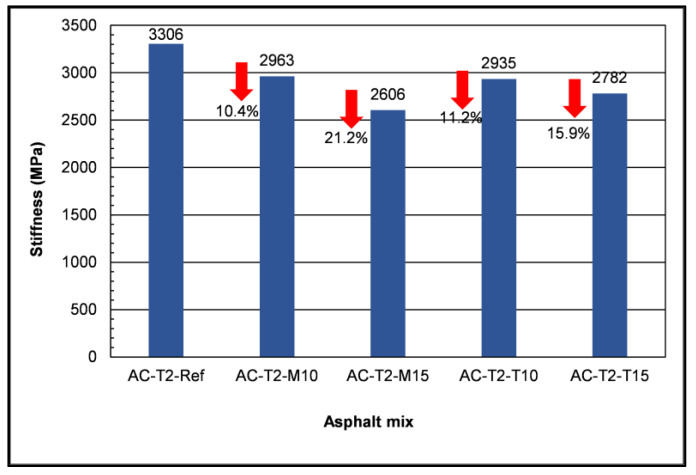
Stiffness results and stiffness reduction with waste glass addition.

**Figure 9 materials-16-04938-f009:**
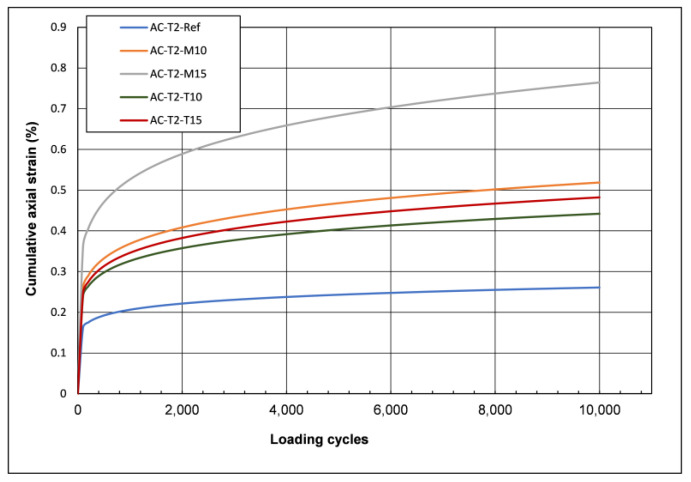
Cyclic compression test results-Cumulative axial strain.

**Figure 10 materials-16-04938-f010:**
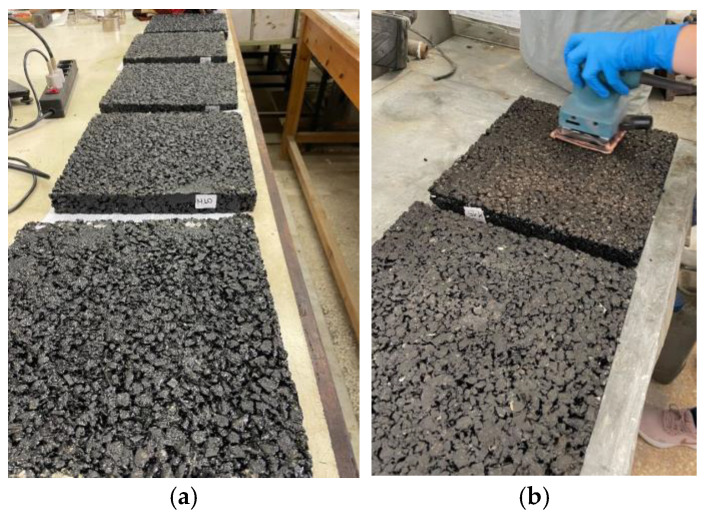
(**a**) Produced slabs for reflectivity. (**b**) Light abrasion of slabs.

**Figure 11 materials-16-04938-f011:**
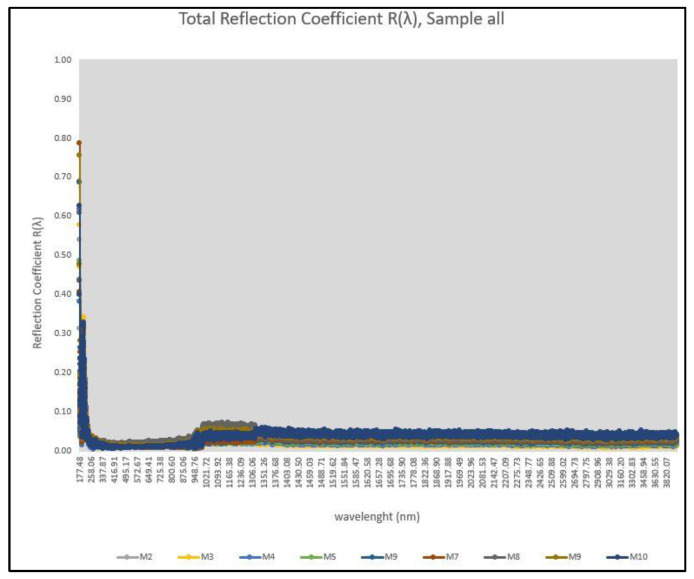
Reflection coefficient for all slabs produced.

**Table 1 materials-16-04938-t001:** Characteristic properties of bitumen.

Property	Specification	Value
Penetration	EN1426 [31]	61
Softening point (°C)	EN1427 [32]	50
Force ductility (J/cm^2^)	EN 13589 [33] & EN 13703 [34]	0.885
Dynamic viscosity (150 °C, Pa·s)	EN 13302 [35]	0.28
Dynamic viscosity (100 °C, Pa·s)	EN 13302 [35]	2.3

**Table 2 materials-16-04938-t002:** Waste glass properties.

Property	Specification	Waste Glass Type	
		Monolithic	Tempered
Flakiness Index (%)	EN 933-03 [37]	20	18
Particle density (Mg/m^3^) ^(1)^	-	2493.7 ±0.01229%	2485.6 ±0.0042%
Shape Index (%)	EN 933-04 [38]	14	13
Freeze–thaw performance (Mass change %)	EN 1367-01 [39] EN 1367-06 [40]	0.2 0.0	0.5 0.5
Micro-Deval (wet method) (M_DE_ for 4.0–6.3 mm glass)	EN 1097-01 [41]	69.8	66.0

^(1)^ Particle density was measured with an electronic gas pycnometer.

**Table 3 materials-16-04938-t003:** Properties of coarse aggregates and limestone sand.

Aggregate Type	Test	Specification	Value
Coarse	Particle density	EN 1097-06 [42]	2882 Mg/m^3^
	Water absorption	EN 1097-06 [42]	0.80%
	Flakiness Index	EN 933-03 [37]	12%
	Los Angeles	EN 1097-02 [43]	20%
	PSV	EN 1097-08 [44]	58%
	AAV	EN 1097-08 (Annex A) [44]	5.5%
Limestone sand	Particle density	EN 1097-06 [42]	2515 Mg/m^3^
	Water absorption	EN 1097-06 [42]	1.20%
	Sand equivalent	EN 933-08 [45]	71%
	Methylene blue	EN 933-09 [46]	1.7 g/kg

**Table 4 materials-16-04938-t004:** AC Type 2 gradation limits.

Sieve Size (mm)	AC Type 2 Gradation Limits (% Passing)
20	100
12.5	90–100
10	60–90
4	18–48
2	5–30
1	4–24
0.25	3–16
0.063	2–8

**Table 5 materials-16-04938-t005:** Contents of asphalt mix constituents.

Asphalt Mix	Aggregate Fraction Content (%)			Glass Content (%)	
	Coarse	Sand	Filler	Monolithic	Tempered
AC Type2-Ref.	83	15	2	-	-
AC Type2–10%-Mon.	75	13	2	10	-
AC Type2–15%-Mon.	75	8	2	15	-
AC Type2–10%-Tem.	75	13	2	-	10
AC Type2–15%-Tem.	75	8	2	-	15

**Table 6 materials-16-04938-t006:** The number of specimens and slabs produced for testing.

Test	Specification	Total Number of Specimens/Slabs
Compactibility	EN12697-10 [47]	10 specimens
Stiffness	EN12697-26 [48]	30 specimens
ITS and ITSR	EN12697-12 [49] & EN12697-23 [50]	30 specimens ^1^
Cyclic Compression test	EN12697-25 [51]	10 specimens
Reflectivity	-	5 slabs

^1^ These are the same specimens used for stiffness testing.

**Table 7 materials-16-04938-t007:** Air void per cent of specimens produced.

No. Specimen	AC-T2-Ref	AC-T2-M10	AC-T2-M15	AC-T2-T10	AC-T2-T15
	Air Voids (%)				
1	11.6	11.9	11.1	12.2	10.6
2	11.9	11.2	10.6	10.3	11.6
3	10.8	12.0	11.0	10.6	10.0
4	11.5	10.9	11.8	11.2	10.1
5	12.0	11.4	10.6	10.9	9.7
6	11.6	11.2	11.5	11.8	11.5
7	11.5	10.8	11.7	11.7	11.5
8	12.0	11.1	10.6	11.7	10.8
Average	11.6	11.3	11.1	11.3	10.7
St. Deviation	0.39	0.44	0.50	0.66	0.75

**Table 8 materials-16-04938-t008:** Compactibility curves equations and parameter K for all AC-T2 mixtures.

Asphalt Mix	Compactibility Curve Equation	Parameter K
AC-T2-Ref	Vα = 21.885 − 3.7151ln(N)	3.7151
AC-T2-M10	Vα = 27.553 − 4.6710ln(N)	4.6710
AC-T2-M15	Vα = 35.819 − 6.0722ln(N)	6.0722
AC-T2-T10	Vα = 26.175 − 4.4374ln(N)	4.4374
AC-T2-T15	Vα = 28.793 − 4.8812ln(N)	4.8812

**Table 9 materials-16-04938-t009:** Stiffness results.

No. Specimen	AC-T2-Ref	AC-T2-M10	AC-T2-M15	AC-T2-T10	AC-T2-T15
	Stiffness (MPa)				
1	3255	2930	2552	3056	2826
2	3175	2935	2598	2958	2874
3	3301	2856	2635	2965	2726
4	3259	2974	2617	2895	2715
5	3425	3000	2665	2889	2803
6	3418	3085	2567	2845	2745
Average	3306	2963	2606	2935	2782
St. Deviation	99	77	42	75	63

**Table 10 materials-16-04938-t010:** Indirect tensile strength and ITSR results.

No. Specimen	AC-T2-Ref	AC-T2-M10	AC-T2-M15	AC-T2-T10	AC-T2-T15
	Indirect Tensile Strength of Dry Specimens (kPa)				
1	942	865	813	882	822
2	935	884	804	873	837
3	927	873	815	870	829
Average	935	874	811	875	829
St. Deviation	7.5	9.5	5.9	6.2	7.5
No. specimen	Indirect Tensile Strength of wet specimens (kPa)				
4	795	698	602	726	651
5	815	705	598	735	643
6	798	715	602	749	657
Average	803	706	601	737	650
St. Deviation	10.8	8.5	2.3	11.6	7.0
ITSR (%)	85.9%	80.8%	74.1%	84.2%	78.4%

**Table 11 materials-16-04938-t011:** Cyclic compression test results.

Asphalt Mix	Cumulative Axial Strain (%)	Creep Rate (μm/m/Loading Cycles)
AC-T2-Ref	0.40	0.08
AC-T2-M10	0.79	0.21
AC-T2-M15	1.16	0.33
AC-T2-T10	0.67	0.16
AC-T2-T15	0.73	0.19

**Table 12 materials-16-04938-t012:** Reflectivity test results.

Asphalt Mix	SRItot (%)	Standard Deviation SRtot * Solar Spectrum	Uncertainty Rtot * Solar Spectrum
AC-T2-Ref	1.621	0.0026	0.0017
AC-T2-M10	1.454	0.0037	0.0023
AC-T2-M15	1.910	0.0024	0.0015
AC-T2-T10	2.076	0.0024	0.0015
AC-T2-T15	1.922	0.0024	0.0015

## Data Availability

Data of this research are available after request.

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
