# Peer review of "Assessment of Waste Glass Incorporation in Asphalt Concrete for Surface Layer Construction"

_materials, 2023, doi:10.3390/ma16144938_

Round 1
Reviewer 1 Report
I congratulate the authors on an interesting work. The purpose and scope of the research project has been clearly described and presented in the work and does not require substantive supplementation. I am only asking you to complete a few elements in the work to increase its readability:
1. Please complete the D50/70 bitumen properties in table 1
2. Figure 3 - complete the zoom level and the type of microscope used for research
3. Please justify why AC grain size up to 12.5 mm was chosen? It is commonly known that a mixture of up to 8 or max 11 mm is used for wearing courses to obtain better macro-texture properties and significantly reduce noise.
4. Please complete table 5 showing the % values in kg/tonne, taking into account the density of the materials
The conclusions resulting from the research are correct and rather not surprising because glass as an addition to the mineral-asphalt mixture for the wearing course is not suitable because it reduces durability. Another barrier is the cost of glass processing and the cost of transport, as well as the troublesome method of dosing in the production process.
Author Response
We thank the Reviewer for his/her comments. We upload our answers to the file attached.

Reviewer 2 Report
This paper reports the investigation results of asphalt concrete mixed with waste glass, monolithic, and tempered at 10 and 15% rates. The authors found that the stiffness moduli of these glassphalts were lower than that of reference asphalt. Based on ITSR, the authors found that Glassphalt mixtures with 15% monolithic or tempered waste glass are not suitable for use in pavement surface layers. They also found no significant difference in SRI values between reference asphalt and these glassphalts. This paper assessed the influence of glass mixture on mechanical and reflection properties, although the materials afforded negative results for practical use. The reviewer thinks the following points should be addressed in the manuscript.
What is the reason that the glass type influenced the mechanical properties? How the difference in shape and size of crushed glass influenced them?
The conditions of the crusher should affect the particle size distributions. How did the authors optimize or choose a condition used in this work?
Please add the scale bar in the panels of Figure 3.
The introduction seems like abstract section. The reviewer suggests modifying the content by removing the “Background” heading.
Author Response

(The authors gave the same response as above.)
